# Theory, Method and Practice of Metal Deformation Instability: A Review

**DOI:** 10.3390/ma16072667

**Published:** 2023-03-27

**Authors:** Miaomiao Wan, Fuguo Li, Kenan Yao, Guizeng Song, Xiaoguang Fan

**Affiliations:** 1State Key Laboratory of Solidification Processing, School of Materials Science and Engineering, Northwestern Polytechnical University, Xi’an 710072, China; 2Shaanxi Key Laboratory of High-Performance Precision Forming Technology and Equipment, Northwestern Polytechnical University, Xi’an 710072, China; 3National Innovation Center of Forging and Ring Rolling Technology in Defense Industry, Northwestern Polytechnical University, Xi’an 710072, China

**Keywords:** metal deformation instability, intrinsic characteristics, structural geometry, theory and methods, engineering application

## Abstract

Deformation instability is a macroscopic and microscopic phenomenon of non-uniformity and unstable deformation of materials under stress loading conditions, and it is affected by the intrinsic characteristics of materials, the structural geometry of materials, stress state and environmental conditions. Whether deformation instability is positive and constructive or negative and destructive, it objectively affects daily life at all times and the deformation instability based on metal-bearing analysis in engineering design has always been the focus of attention. Currently, the literature on deformation instability in review papers mainly focuses on the theoretical analysis of deformation instability (instability criteria). However, there are a limited number of papers that comprehensively classify and review the subject from the perspectives of material characteristic response, geometric structure response, analysis method and engineering application. Therefore, this paper aims to provide a comprehensive review of the existing literature on metal deformation instability, covering its fundamental principles, analytical methods, and engineering practices. The phenomenon and definition of deformation instability, the principle and viewpoint of deformation instability, the theoretical analysis, experimental research and simulation calculation of deformation instability, and the engineering application and prospect of deformation instability are described. This will provide a reference for metal bearing analysis and deformation instability design according to material deformation instability, structural deformation instability and localization conditions of deformation instability, etc. From the perspective of practical engineering applications, regarding the key problems in researching deformation instability, using reverse thinking to deduce and analyze the characteristics of deformation instability is the main trend of future research.

## 1. Introduction

In the majority of engineering application fields, such as aerospace, automobile, coastal power station, etc., the deformation and stability failure of materials is one of the key issues. From the perspective of macrophysics, instability is a phenomenon in which a structure changes from one equilibrium state to another equilibrium state, or to an unstable state. In essence, instability is a change in energy. The deformation instability of materials not only has macroscopic instability, but also has microscopic instability [1]. Generally, the deformation instability of the material itself or its structure is related to stiffness, and shearing is considered to be the most fundamental cause of deformation instability [2]. Plastic deformation instability occurs during the deformation process of metal materials. If the material properties, loading conditions, and geometric structures are different, the principal stress states suffered are different in plastic instability. The deformation instability of metal materials will be manifested in the form of local necking [3,4,5], buckling [6,7,8], wrinkling [9,10,11], etc. Since there are different process parameters, such as forming temperature, friction conditions, forming equipment, process route, etc., the change law of the same instability characteristic is also different.

Therefore, deformation instability is a process based on energy change and guided by shearing deformation, and the deformation instability process of metal materials is a complex process with multi-factor coupling. That is, it is mainly related to multiple factors, such as material intrinsic properties, material macro/microstructure, geometric structure, ambient temperature, and loading conditions [12,13,14,15,16]. In practical engineering applications, it is sometimes necessary to take advantage of deformation instability. That is, some parts may need to use the deformation instability characteristics of materials to achieve special functions, and then obtain high-efficiency applications in specific occasions. Sometimes, the deformation instability of materials may bring potential danger to the service of parts, or directly cause catastrophic accidents, and it is necessary to accurately predict and precisely control the occurrence and development of deformation instability. Therefore, the study of deformation instability is very important for the development of precise plastic forming for high-performance components and better applications in engineering practice.

At present, many scholars have adopted theoretical analysis [7,17], numerical simulation [18,19,20,21] and experimental research methods [22,23]; there is systematic research on deformation instability in terms of materials, structure, boundary conditions and applications. That is, the research on the deformation instability of metal materials first needs to determine the instability behavior, establish the mathematical model, and obtain the analytical solution of deformation instability. Secondly, combined with the mathematical model and numerical simulation, stress–strain analysis is carried out to determine the type of deformation instability. Finally, by combining numerical simulation and experimental research methods, the variation law of the deformation instability is analyzed and the universal criterion and optimal process parameters which are suitable for specific engineering fields are obtained. According to the above, this paper first makes a comparative analysis and summary for the definition of deformation instability, classical instability criteria and the principle of deformation instability, and then the latest research on instability criteria in recent years is summarized. Secondly, in terms of the material inherent characteristics and boundary conditions, the super-plasticity and deformation instability characteristics of metal materials under hot forming conditions are discussed. The deformation instability modes of different structures are also described by classifying them into metal sheets, tubes and beams, respectively. Since there are many research methods on the deformation instability of metal materials, the research methods in the latest literature are reviewed and compared. Finally, they are classified according to the effect of deformation instability in engineering applications; that is, deformation instability is positive and constructive, or deformation instability is negative and destructive. Therefore, we further summarized the latest literature on deformation instability in practical engineering applications and proposed the unresolved problems and future research directions of deformation instability.

Up until now, there have been many review papers on deformation instability, most of which only focused on the theoretical analysis of deformation instability (instability criteria); there have been very few papers that classify and review based on material characteristic response, geometric structure response, analysis methods and engineering applications. Therefore, from the above four aspects, this paper comprehensively analyzes, summaries and reviews the latest discoveries and achievements in the study of the deformation instability of metal materials. The purpose is to help more researchers quickly extract the latest research results in this field, and compare the value and significance of their own topics with predecessors, so as to propose more innovative research content.

## 2. Definition of Deformation Instability

Since deformation instability is closely related to the intrinsic characteristics of materials, macro/microstructure, forming temperature, stress loading conditions and other factors, many scholars have clearly defined deformation instability for different materials and different forming conditions. At the same time, relevant theoretical criteria are established to predict, analyze and control the deformation instability of materials.

Based on the problem of elastic compression rods, Euler proposed the definition of compression rod instability first; that is, the minimum load which causes the compression rod to bend is the critical load of compression rod instability. It was concluded that critical load is inversely proportional to the square of rod length (as show in Equation (1)), also known as the “engineering beam theory”. Since then, people have had an understanding of deformation instability. Felippa et al. [24] further interpreted and reasoned the formula of instability deformation of beam structure, and explained it in detail. Abrahamson [25] proposed the dynamic buckling theory of rods under axial compression, which laid the foundation for subsequent research on deformation instability.
(1)$$P_{cr}=\frac{EI\pi^{2}}{l^{2}}$$

When the majority of metal materials undergo plastic deformation, the essence is also a change of energy. When the increment of strain energy is less than the increment of work performed by an external force, plastic instability will occur. From the perspective of stress and loading force, plastic instability refers to the phenomenon that when the load on the material reaches a certain critical value, even if the load decreases, the plastic deformation will continue. Plastic deformation instability includes load instability and deformation instability. For general metal materials, load instability is accompanied by geometric deformation instability; the material will be unstable when there are significant extreme points on the tensile curve. In a deformation process dominated by tensile stress, deformation instability is mainly manifested in the form of shear deformation and local necking, which is called tensile instability [26]. On the other hand, in a deformation process dominated by compressive stress, deformation instability is mainly present in the form of wrinkling, buckling, shear fracture, etc., which is called compression instability [27] (as show in Figure 1). It also can be seen that deformation instability is closely related to the bearing stress state.

Tensile instability includes dispersion instability and concentration instability. After the material undergoes stable and uniform deformation under tensile stress, there is a metastable flow in a relatively wide area, which is called dispersion instability. When the tensile necking reaches a certain extent, the unstable flow will be confined to a narrow area, which is called local instability or concentration instability. Swift proposed [28] that, for ductile work-hardening materials with uniform structure, the strain remains constant until a critical value is reached. At this point, the rate of decrease in the cross-sectional area is faster than the rate of increase in yield stress, causing the material to reach its maximum load capacity. This indicates the occurrence of load instability, which means that the strain is no longer uniformly distributed throughout the material. As the load decreases, localized necking continues and dispersion instability in materials occurs. Therefore, Swift proposed a dispersion instability theory for uniaxial tensile, where instability conditions are as follows:(2)$$\frac{d\sigma }{\sigma }=d\varepsilon =-\frac{dA}{A}$$
where dσ is stress increment, dε is strain increment, *A* is instantaneous cross-sectional area, and *dA* is the value of the change in cross-sectional area.

In the complex stress state, the stress intensity $\sigma_{i}$ and strain intensity $\varepsilon_{i}$ reflect the comprehensive effect of each stress–strain component. Therefore, the dispersion instability condition can also be written as:(3)$$\frac{d\sigma_{i}}{\sigma_{i}}=d\varepsilon_{i}$$

If the stress–strain relationship of materials follows the power function form $\sigma_{i}=K\varepsilon_{i}^{n}$, when unidirectional tensile, using dispersion instability condition $\sigma_{i}=\sigma_{1},\;\varepsilon_{i}=\varepsilon_{1}$, the strain intensity at dispersion instability can be expressed as:(4)$$\varepsilon_{i}=\varepsilon_{1}=n$$
where *K* and *n* are material constants.

Hill proposed the definition of concentration instability in 1952, firstly [29], which refers to the generation and development of concentration instability necking, mainly dependent on the local thinning of a sheet without any length change occurring along the direction of the thin neck. As a result, the condition for instability is that the strengthening rate of the materials’ unstable section is mutually balanced with the reduction rate in the thickness direction. At this moment, the local necking may continue to develop, while the stress in other parts will remain constant or even decrease, ultimately leading to the cessation of deformation. Hill analyzed the local necking problem of the Luders line at the stress field of $\sigma_{2}<\left(1/2\right)\sigma_{1}$ in thin plate. Suppose that the direction of characteristic line is *y*, the direction perpendicular to it is *x*, and the angle between maximum principal stress $\sigma_{1}$ and *x* is α. The local instability criterion derived by Hill is as follows:(5)$$\frac{d\sigma_{x}}{\sigma_{x}}=\frac{d\sigma_{y}}{\sigma_{y}}=-d\varepsilon_{t},\;d\varepsilon_{y}=0$$

The direction of local necking line is:(6)$$\tan^{2}\alpha =-\frac{d\varepsilon_{1}}{d\varepsilon_{2}}=-\frac{\sigma_{1}-\frac{1}{2}\sigma_{2}}{\sigma_{2}-\frac{1}{2}\sigma_{1}}$$

Marciniak and Kuczynski jointly proposed the famous M–K groove theory [30]. According to the M–K theory, deformation instability occurs in a groove perpendicular to the direction of larger principal stress, which leads to the gradual concentration of local strains. In the initial stage, the change in groove depth is associated with a gradual decrease of the strain in an adjacent area. When the strain is reduced to a certain limit value, $\varepsilon^{*}$, sheet tensile instability occurs. The schematic diagram of the classical M–K theory is shown in Figure 2a. M–K theory is commonly employed to predict local necking phenomena or evaluate forming limit strain, and it is a common method to calculate the forming limit curve (FLC) for local necking strain in thin sheet [31,32]. However, it should be noted that the classical M–K model assumes that the initial groove is perpendicular to the principal stress direction, so it is only suitable for calculating the ultimate strain in the right region of FLC [33]. In order to predict the ultimate strain of the metal sheet in the left region of FLC, Hutchinson and Neale tilted the assumed initial grooves in the model, by which the M–K model was modified to form a certain angle ψ with the secondary stress direction [34]. Figure 2b shows the schematic diagram of the modified M–K model, and the instability criterion is as follows:(7)$$\tan\psi =\exp\left[\left(1-\rho \right)\varepsilon_{11}\right]\tan\psi_{0}$$
where ψ is the inclination angle of current groove, $\psi_{0}$ is initial inclination angle, ρ is strain path, and $\varepsilon_{11}$ is principal strain.

In recent years, numerous researchers have modified the classic M–K instability theory for different materials, allowing for the prediction of deformation instability and sheet forming limits under different forming conditions. A summary of the research results can be found in Table 1.

Storen and Rice [48] were able to predict the occurrence of localized necking in a sheet subjected to biaxial tension using a simple constitutive model with only one vertex at the subsequent yield point. It was found that if the plastic flow theory of ideal yield locus was adopted, the uniform thin sheet could not appear with local necking under the biaxial tension action; that is, the sheet deformation would be stable. However, the ideal yield trajectory cannot accurately describe the deformation process in the deformation strengthening stage. This is because, in the subsequent yielding stage of the deformation process, singularity points must exist on the yield surface, resulting in regions of increased local stress [49]. Therefore, Storen and Rice believed that the deformation instability of materials is induced by the presence of singularity on the subsequent yield surface. The singularity point appears because of the orthogonal slip inside the material crystal, which makes the yield surface of the polycrystalline material form an irregular apex during the plastic deformation, and this singular apex causes the local necking of the materials. The plastic flow principle of ideal yield surface is shown in Figure 3a, and the principal stress rate at point A is expressed as:(8)$$\left\{ \begin{array}{l} {\dot{\varepsilon }}_{1}=\frac{1}{2h}m\left(m:\dot{\sigma }\right)\\ {\dot{\varepsilon }}_{1}=\frac{1}{2h}m\left(\dot{\sigma }-m:\dot{\sigma }\right) \end{array}\right.$$
where **m** is normal vector of the yield surface and *h* is the hardening rate.

Figure 3b is a schematic diagram of plastic flow with a singularity point, where the two yield surfaces intersect at point B. Therefore, the strain rate at point B is expressed as:(9)$$\dot{\varepsilon }=\frac{1}{2h}\left[m_{1}\left(m_{1}:\dot{\sigma }\right)+m_{2}\left(m_{2}:\dot{\sigma }\right)\right]$$

This research shows that, compared with the classic M–K model, the Storen–Rice model predicts forming limit diagrams for an aluminum alloy sheet in biaxial tension; it is closer to the experimental results [50,51].

Bressan and Williams [52] studied the shear instability and local necking in sheet metal forming. They concluded that the essence of plastic deformation instability is the change of energy, and deformation instability is controlled by shearing. Work hardening and an inhomogeneity of materials can promote the development of strong shear bands in a pure shear direction. The primary reason for the development of shear bands, which in turn leads to the shear instability of materials, is the growth of voids within the materials [53].

Semiatin et al. [2,54] analyzed the adiabatic shear bands of metallic materials based on load instability and local fluidization models. According to the local fluidization model, significant strain concentrations only occur in simple shear if the deformation exceeds instability strain. A flow localization model of strain rate hardening and thermal softening effects can be used to represent the dynamics of strain concentration, and thus it can predict the occurrence of adiabatic shear banding.

Many scholars have given corresponding definitions of deformation instability according to different materials’ formability. For example, Song et al. [55,56] studied the forming properties of superplastic materials, and proposed that the occurrence of materials’ deformation instability is attributable to the local narrow necking, and the expansion phenomenon of necking regions in defective materials when stress is loaded; it also experienced three processes of load instability, geometric instability and fracture instability. Load instability occurs when the tensile load increases by d*P* ≤ 0. Geometric instability is the process of materials’ non-uniform deformation; that is, when the uniform strain is greater than the strain hardening exponent, the strain hardening effect cannot offset the increase in stress caused by the contraction of cross-section, and geometric instability occurs at this time. Typically, in the plastic instability process, load instability and geometric instability usually occur simultaneously. In the case of superplastic materials, necking does not occur after load instability, and it can re-establish a prolonged and nearly stable deformation process after geometric instability [57]. In addition, Li et al. [58,59] found that plastic deformation instability is related to material scale by studying the plastic deformation instability of Au/Cu multilayer film. At nanoscale, the dislocations between grains and crossings at the grain interface can easily cause local shear bands to appear in the material, ultimately resulting in geometric instability. At higher scales, localized dislocations within the material displace the grain boundary interface, which also result in plastic deformation instability.

Based on the above definitions of deformation instability, in order to more accurately analyze and predict the deformation instability behavior of materials, a variety of deformation instability criteria have been used for judgment and prediction. Among them, several common criteria are as follows:(1)Hart’s instability criterion [60]

Under uniaxial tension, the material appears to have deformation instability when the following conditions are met:(10)γ+m<1

γ is the work hardening rate:(11)γ=(1/σ)(∂σ/∂ε)

*m* is the strain rate sensitivity parameter:(12)$$m=\left(\dot{\varepsilon }/\sigma \right)\left(\partial \sigma /\partial \dot{\varepsilon }\right)=\partial \ln\sigma /\partial \ln\dot{\varepsilon }$$
where σ is the flow stress, ε is real strain, and $\dot{\varepsilon }$ is real strain rate.

(2)Jona’s instability criterion [61]

Under uniaxial compression, the localized flow criterion is as follows:(13)$$\xi_{1}=\gamma +m-1>0$$

According to the above criterion, flow localization increases with the increment of material strain rate sensitivity. Taking into account the localization of strain rate, an additional criterion given for flow localization in uniaxial compression is as follows:(14)$$\xi_{2}=\frac{\gamma -1}{m}>0$$

(3)Semiatin’s instability criterion [62]

Semiatin and Lahoti proposed a phenomenological criterion for predicting flow localization in the hot forging of titanium alloys. According to this criterion, shear banding may occur when the ratio of dimensionless flow softening rate to strain rate sensitivity parameter exceeds 5. The instability condition for uniaxial compression can be expressed as:(15)$$\alpha =-\frac{\gamma }{m}>5$$
where α is the flow localization parameter.

(4)Dynamic Material Model (DMM) criterion [63]

The DMM is based on the principle of irreversible thermodynamics proposed by Ziegler, which is suitable for large deformation processes of metallic materials. When the entropy rate produced by the material deformation does not match the entropy rate imposed on the material, flow localization or instability will occur. Therefore, the flow instability criterion is:(16)$$\frac{\partial D}{\partial \dot{\varepsilon }}<\frac{D}{\dot{\varepsilon }}$$
where *D* is the dissipation function that characterizes the flow behavior of the material, and $\dot{\varepsilon }$ is strain rate. As suggested by Kumar and Prasad, if power is divided by two parts *G* and *J*, *D* can be replaced by *J*, this is called power co-content.
(17)$$J=\int \dot{\varepsilon }d\sigma$$

Therefore, the instability criterion also can be expressed as follows:(18)$$\frac{\partial J}{\partial \dot{\varepsilon }}<\frac{J}{\dot{\varepsilon }}$$
when the material obeys the following constitutive equation:(19)$$\sigma =K{\dot{\varepsilon }}^{m}$$
(20)$$J=\frac{m\sigma \dot{\varepsilon }}{m+1}$$

Substituting the expression of *J* into the instability criterion, the following expression can be obtained:(21)$$\xi_{3}\left(\dot{\varepsilon }\right)=\frac{\partial \ln\left[m/\left(m+1\right)\right]}{\partial \ln\dot{\varepsilon }}+m\leq 0$$

According to the continuity criterion, when the parameter $\xi_{3}$ becomes negative, the flow becomes unstable.

Based on the DMM criterion, the formability and stability of different materials under high temperature deformation conditions can be accurately analyzed by the forming diagrams method. Chen et al. [64] established a dynamic material model by using isothermal compression experiments to study the thermal deformation behavior of T2 copper. According to the forming diagram, it was found that plastic flow instability mainly occurs at low temperatures of 500–650 °C and a strain rate greater than 0.1 s^−1^. Zheng et al. [65] studied the high-temperature formability of high-strength Mg alloys through tensile experiments. The forming diagram of the Mg alloy was designed based on DMM, and it was found that, with the decrease of the strain rate and the increase of deformation temperature, the fracture instability patterns exhibited by the tensile specimens vary from quasi-cleavage to ductile fractures. For aluminum alloys, the tensile specimens’ fracture process is from brittle to ductile [66]. Liu et al. [67] researched the formability of cast steel by forming a graph method of DMM; it was found that, with the increase of temperature and the decrease of strain rate, the degree of dynamic recrystallization increases, and deformation instability is prone to occur.
(5)Gegel’s and Alexander’s instability criterion [68]
(22)0<m≤1
(23)$$\frac{\partial \eta }{\partial \left(\ln\dot{\varepsilon }\right)}<0$$
(24)$$s=\frac{\partial \log\sigma }{\partial \left(\frac{1}{T}\right)}\geq 1$$
(25)$$\frac{\partial s}{\partial \left(\ln\dot{\varepsilon }\right)}<0$$
where *m* is the strain rate sensitivity, *s* is the entropy, and *T* is temperature. Meanwhile, considering the Lyaponov function L(η, s), Alexander proposed some conditions for the materials’ stable flow. In addition to the above equations, Alexander’s criterion include:(26)$$\frac{\partial m}{\partial \left(\ln\dot{\varepsilon }\right)}<0$$

Both Gegel and Alexander’s flow stability criteria show that the flow stress with respect to strain rate curve should be inherently convex, and should exhibit flow softening with incremental temperature within materials. Furthermore, according to Alexander’s criterion, strain hardening should decrease with increment strain rate and increase with increment temperature.

(6)Metallurgical instability criterion [69]


(27)
2m<η≤0



(28)
$$\eta =\frac{2m}{m+1}$$


Therefore, the metallurgical instability condition is m<0.

In summary, the definition and criteria of deformation instability under specific conditions are given according to the deformation instability characteristics of different materials, different structures, and different deformation conditions. Deformation instability is a process based on energy change and guided by shearing deformation. Deformation instability will promote or inhibit the effective forming of materials, which is very critical for engineering applications.

## 3. Deformation Instability Induced by Characteristics of Material

Deformation instability is related to material characteristics; whether it is the inherent characteristics of materials or the new characteristics induced by the change of external conditions, the deformation instability characteristics will be different. Therefore, this section mainly reviews the deformation instability characteristics induced by inherent properties of materials, and the deformation instability characteristics of new properties, which are induced by changes in the forming conditions.

### 3.1. Deformation Instability in Superplastics of Materials

Different materials have different inherent characteristics. It is well known that super-plasticity is the inherent characteristic of superplastic materials, and the super-plasticity of materials can also be a new characteristic induced by high temperature, multi-pass processing and other external conditions. In general, super-plasticity is an underlying property of materials with lattice structures. From the perspective of bearing capacity, superplastic materials do not show geometric instability after load instability, and there will be a similar stable deformation process after geometric instability [56]. Microscopically, superplastic behavior is mainly divided into microcrystalline super-plasticity and microstructural super-plasticity. It is characterized by relatively low flow stress. The failure mode of superplastic materials is dominated by unstable plastic flow, and it shows the uniform strain before deformation instability [70,71,72]. Its microstructure shows certain void characteristics.

Zhang et al. [73] established the superplastic limit diagram of polycrystalline superplastic material; it was found that the strain rate sensitivity index *m* and strain hardening exponent *n* are the main mechanical parameters which affect the deformation instability of superplastic material; the deformation instability mode is mainly represented by necking at high strain rate. Sheinerman [74] proposed a model to describe the occurrence of single and multiple necks in super-plastically deformed materials; it was found that diffuse necking occurs throughout the specimen in small samples of super-plastically deformed ultrafine-grained metals and alloys. In larger specimens, the instability region was only in a small portion of the sample, which can significantly reduce deformation failure. Song et al. [55,56] analyzed the influence of mechanical parameters and temperature on the deformation instability of superplastic materials; they found that deformation instability is closely related to the strain rate sensitivity index *m*, strain hardening exponent *n*, and hardening coefficient. They put forward the criterion of tensile load instability and geometric instability under constant temperature; that is:(1)Load instability criterion are as follows:
(29)$$\gamma \leq 1+m\;\mathrm{or}\;\varepsilon \geq \frac{n}{1+m}$$

According to the load instability criterion, the load instability point of superplastic materials appears earlier than plastic materials.

(2)The geometric instability criterion is as follows:


(30)
$$\gamma \leq 1-m\;\mathrm{or}\;\varepsilon \geq \frac{n}{1-m}$$


Stress instability is manifested at this point because material necks occur when the materials have geometric instability.

In addition, a majority of metal materials will also exhibit superplastic deformation behavior under the action of high temperature, fine-grain strengthening, or other special conditions. Jafarian et al. [75,76] studied the superplastic behavior of ultrafine-grained aluminum alloys, and found that the occurrence of dynamic recrystallization destabilized superplastic deformation. Meanwhile, the low-angle grain boundary gradually transforms into a high-angle grain boundary, and grain boundary slip is the main mechanism of superplastic deformation instability (as shown in Figure 4a). The condition for superplastic deformation instability is $T>300\;℃,\;\dot{\varepsilon }\;>5\times 10^{-2}$. Li et al. [77] conducted hot rolling and heat treatment on the fine-grained(average grain size is 8.48 μm) 5A70 aluminum alloy sheet, and they found a phenomenon of fine precipitates dispersed during the superplastic deformation process, as shown in Figure 4b. The presence of these microscopically dispersed particles promotes the nucleation and growth of voids, ultimately leading to the instability of superplastic deformation. 

Malik et al. [78] studied the superplastic instability behavior of fine-grained Mg alloys (as shown in Figure 5). It was found that the grain size increased significantly after superplastic deformation in high temperature tensile tests. By analyzing the stress–strain curves at different temperatures and deformation rates, it was found that the mechanism of superplastic deformation instability was intragranular slip because of the small value of significant strain rate sensitivity (*m*) and hardening exponent (*n*). Nazeer et al. [79] also suggested that, for a very fine grain size which has uniform microstructure, the conditions of high *m*, low *n*, and thermal stability are necessary to achieve super-plasticity. It was found that the elongation of the WE54 Mg alloy reached 726% at 400 °C. If this temperature is exceeded, the thermal stability of the material will decrease, the super-plasticity will deteriorate, and deformation instability and cracks will even be generated, thus limiting the further extension of the material and eventually leading to deformation failure. Recent studies by some scholars on the characteristics of the superplastic deformation instability of other alloy materials are summarized in Table 2.

In order to meet the requirements of lightweight materials, and better performance in aerospace, automotive and other industrial fields, research and development of various new materials are increasing gradually; they are made of various materials. Due to the different functions of various materials, functionally gradient metal materials (FGMM) have been formed [84,85]. FGMM are generally composed of metal and another brittle material, which has the advantages of corrosion resistance, high strength, and the ability to meet the surface stress continuity of composite materials [86,87]. In addition, in the study of formability of FGMM, the instability mechanism research on plastic deformation is also crucial. Tang et al. [88] investigated the superplastic behavior of shear band deflection in soft and hard functionally gradient metallic glasses (GMG). The deformation instability of homogeneous metallic glasses is characterized by the formation of localized shear bands, which can lead to severe damage or macroscopic brittle fractures of the material. The deformation instability mechanism of functionally gradient metallic glasses is shown in Figure 6a. For hard-shell soft-core materials, the main shear band starts from the upper left surface under the bidirectional compressive stress, and the effective shear yield stress of local hard zone gradually increases. The shear force gradually develops towards the soft core area, and the shear angle is largest when it reaches central area. Then, it develops to the hard zone, so that the shear band is deflected. Similarly, for soft-shell hard-core materials, the deflection of the shear band also occurs under the action of bidirectional compressive stress, and shear velocity is faster than hard-shell soft-core materials in the local hard area. The correctness of the principle is verified again through the analysis of fracture morphology obtained in the experiment. Nguyen et al. [89] discussed the bending and buckling of thin-walled sandwich I-beams by considering two types of material distribution, i.e., top-bottom half distribution and enveloping distribution (as shown in Figure 6b). They proposed a modified gradient beam theory and then analyzed the bending and buckling of thin-walled functionally gradient sandwich I-beams on two-parameter elastic foundations by separating the variables.

### 3.2. Deformation Instability in Hot Forming Process

Under thermoforming conditions, the plastic deformation instability of material is characterized by flow instability, and the material flow occurs before local necking. From the analysis of external forming conditions, conditions such as tensile load, friction conditions, temperature gradients or local softening of materials may cause flow instability of materials [2,90]. From the analysis of physical mechanisms, many factors such as adiabatic shear band, dynamic recrystallization, grain coarsening or spheroidization can lead to material structure instability and flow instability [91,92]. Culver [93] concluded that thermal softening counteracts the stability that was produced by work hardening during hot forming, and this phenomenon may lead to material instability. When the instability condition is reached, the deformation becomes extremely localized and localization occurs in pure shear. This localization can concentrate in a single shear plane, eventually leading to localized necking of the material. According to different forming methods, this can be roughly divided into thermal compression instability and thermal tensile instability.

Ji et al. [94] investigated the deformation instability behavior of the Ti-6Al-4V alloy in the thermal compression process by combining experimental studies and numerical simulation methods, taking bolt forging and forming as an example. After analyzing the macro-instability index and forming load curve, it was found that the bottom of the bolt head firstly presents shear instability. This can be attributed to the elevated local temperature, causing a reduction in the deformation load due to localized softening. Eventually, this leads to a macroscopic bolt instability. From the analysis of processing diagrams at different strain rates and different temperatures, instability regions appear in the region of smaller strain rates and low temperatures, or at high temperatures and high strain rates (as shown in Figure 7a). As a result, the strain rate at the bottom of bolt head is low and the temperature is high in forging process, thus increasing the likelihood of macroscopic deformation instability. From the analysis of the microscopic mechanism, the Ti-6Al-4V alloy is most prone to microstructure instability under conditions of high temperature and high strain rate, because high strain rate leads to the formation of an adiabatic shear band or flow localization at low temperatures, and intergranular deformation at high temperatures to crack [95]. Ma et al. [96] and Ling et al. [97] discussed the deformation instability behaviors of dual-phase Mg-9Li-3Al and Al-4.96Cu-0.96Mg-0.63Ag-0.57Mn-0.13Zr alloys by thermal compression experiments. Processing diagrams of power dissipation and instability regions were established by using the dynamic material model, as shown in Figure 7b. It was concluded that flow instability regions are located in regions of low temperature and high strain rate. Singh et al. [98,99] conducted uniaxial compression tests on (Nb + V) stabilized micro-alloyed steel with mechanical thermal simulation. From Figure 7c, it has been discovered that, at high strain rate (10 s^−1^), flow instability occurs in both thermal and warm deformation regions. At low strain rates (0.01 s^−1^), flow instability occurs only in lower temperature ranges (700~900 °C). The reason for this is that low strain rates can cause microcracks or micro-void nucleation, leading to instability, while high strain rates can result in the formation of adiabatic shear bands within high temperature regions, ultimately causing shear instability within the materials.

Zheng et al. [65,100] and Imran et al. [101] studied deformation instability characteristics of Mg alloy and TC4 alloy under hot stretching conditions, respectively. The thermal processing diagram is shown in Figure 8a; the MgNdZnZr alloy is unstable in a range of 200–300 °C and 0.02~1 s^−1^. The TC4 alloy is unstable in a range of 800~850 °C and 0.01~1.0 s^−1^. Macroscopic fracture morphology is shown in Figure 8b; it was found that, with temperature increases, the fractures have a characteristic that transitions from a quasi-cleavage plane to a ductile fracture. This is because the temperature increase leads to the homogenization of internal tissues and stress relaxation [102]. Sun et al. [103] analyzed the deformation instability characteristics of a CrMnFeNi high-entropy alloy cast and forged under high temperature tensile conditions from the perspective of physical mechanisms, as show in Figure 8c. It was found that the fracture instability of as-cast parts shifted from ductile fracture to brittle fracture as the parts reached moderate temperatures. At room temperature for the forged parts, the physical mechanism of instability was attributed to slip and twinning, and dynamic instability occurred when the temperature reached 950 °C, causing immediate softening after yielding.

Except for the above contents, the latest research on deformation instability in hot forming by some scholars is summarized in Table 3.

## 4. Deformation Instability Induced by the Structural Geometry of Materials

The deformation instability characteristics of materials are closely related to their macro/micro structures. Many scholars have studied the influence of macro geometric configurations such as sheet metal, tubes, and beam-column structures on deformation instability behaviors. They have good engineering application orientations and provide references for practical applications. Therefore, this section reviews the latest research results on deformation instability characteristics according to different geometric structures of materials.

### 4.1. Deformation Instability of Sheet Metal

The deformation behavior in sheet metal forming processes is relatively complex. In sheet metal stretch forming, deformation instability is mainly in the form of shearing; in the extrusion or stamping forming, the deformation instability is mainly in the form of wrinkling. Shear instability and wrinkling can both cause many problems, such as easy corrosion, easy damage, and difficult assembly [114]. Therefore, it is necessary to avoid deformation instability in most cases of practical applications. Many scholars have studied the influence of non-material intrinsic characteristic parameters such as loading conditions, process parameters, and geometric parameters on deformation instability. Using the method of establishing theoretical criteria, wrinkling limit diagrams (WLD), etc., it is possible to efficiently predict and accurately control the occurrence of sheet metal instability [115,116].

In order to study the local deformation instability characteristics caused by local uneven plastic deformation during the plastic forming process of thin-walled metal sheets, Li R. et al. [117] and Li F. et al. [118] used different types of notched specimens for tensile tests, as shown in Figure 9a. Li R. studied the deformation instability of a 2219-O aluminum alloy sheet in pure shear, dog bone, and curved specimens by experiments. Li F. simulated and analyzed the deformation instability characteristics of V-notch, plane tensile and shear specimens of magnesium alloy sheets. It was found that stress and strain concentration occurred during the stretching process, and they were all concentrated at the root of notch, resulting in a critical area at the edge and center of notch, with localized necking in this area. Du et al. [115,119] used the method of establishing WLD to predict wrinkling in a uniform sheet under symmetrical stretching. It was found that when the thin sheet was stretched symmetrically at both ends, a wrinkle wave along the stretching direction was generated in middle. There is a law that the wrinkle wave increases with the increase of height of the middle wrinkle, and new wrinkle waves are generated on both sides, as shown in Figure 9b. According to the WLD of the thin-walled sheet it was concluded that the corresponding area below the unified critical wrinkling limit curve is the wrinkling area, and the corresponding area above the curve is the wrinkling-free area. Tang et al. [120] studied the wrinkling behavior of a thin-walled sheet under asymmetric stretching, which is basically the same as that under symmetric stretching. That is, the direction of wrinkle is still along the direction of the principal stress, and the wrinkle waves expand to both sides, as shown in Figure 9c.

Chen et al. [121,122] investigated the wrinkling behavior of high curvature, large flange sheet metal stampings by experiment. They found that if the blank holder was not used, the larger the gap between mold and side blank, and the more serious the wrinkling of the flange. However, when the rubber blank holder was used, wrinkling was significantly improved, as shown in Figure 10a. Won et al. [123] proposed a phenomenological for predicting wrinkling; that is, using critical compressive strain and geometric bending strain with respect to a specific triaxiality to predict wrinkling in Gpa-grade steels. Their study revealed that severe compression or folding of the mold surface usually causes wrinkling in the non-flange area, while wrinkling in the flange area is caused by uniaxial compressive strain and buckling strain provided by sufficient blank holder force, as shown in Figure 10b. López-Fernández et al. [124,125] investigated the formability and failure modes within the limit diagram (FLD) of a shrinking flange on the AA2024-T3 sheet by using the single-point incremental forming method (SPIF). In the case of the flange radius being large, there are two modes of wrinkling failure and initial wrinkling failure, as shown in Figure 10c. When the mold radius is small, only initial wrinkling failure occurs; for the SPIF method, when the compressive stress reaches a certain critical value, shrinkage flanging can cause wrinkling.

### 4.2. Deformation Instability of Tubes

Wrinkling is the main form of instability in the tube forming process, and includes bending, compression or liquid expansion. This wrinkling instability generally tends to be localized, and can possibly lead to catastrophic failure such as collapse or rupture [126,127]; wrinkling can also destabilize the tube surface, resulting in thinner or thicker walls [128,129,130,131].

Jia et al. [132,133] studied the bending formability of composite thin-walled lenticular tubes (CTLTS); it was found that CTLTS first showed local wrinkling, and gradually formed periodic wrinkling as the bending angle increased. Finally, the concentrated deformation in the local wrinkled area caused a collapse of the composite thin-walled tube, as shown in Figure 11a. Naderi et al. [134] explored the influence of geometric parameters on the wrinkling of composite tubes, and found that when the thickness variation ratio of composite tubes was less than 80 mm, the probability of wrinkling on the bending inner side increased. Zhu et al. [135,136] investigated the influence of geometric dimensions and filling conditions on the formability of thin-walled composite tubes. When the rigid mandrel filling and the relative bending radius was in a range of 1.32~1.80, the forming defects of thin-walled composite tubes mainly had wrinkling, while when the relative bending radius gradually increased, the outside wall of tubes suffered from cracking failure.

Li et al. [137,138,139,140,141,142] conducted a lot of research on the wrinkling instability of composite bending tubes, thin-walled round bending tubes, and rectangular bending tubes. The relevant research is basically mature, and the research conclusions have very important reference significance. The studies primarily focused on investigating the effects of geometrical parameters, filling conditions, loading conditions and forming parameters on tube wrinkling. The main conclusion was that using hard polymer filling can significantly inhibit tube wrinkling, and the suppression of the inner tube is more significant. However, when rigid constraints are employed, wrinkles appear easily in the core area, as shown in Figure 11b [137,141], and wrinkling occurs more easily under tensile stress than under compressive stress [138]. In the cold-formed condition, the corners can reduce the corrugation height of inner flange, but increase the corrugation height of the sidewall [139]. For multitool constrained bending with different tube shapes, geometric specifications and loading conditions, the wrinkling occurs in different locations, including front, straight, curved, integral, upper and lower bends, etc., but the wrinkling form in bent tubes is similar to regular waves, as shown in Figure 11c [140].

In addition to the rotary draw bending method, there are also other forming methods for the bending tube, such as push bending, free bending, etc. The deformation instability during the forming process is summarized in Table 4.

### 4.3. Deformation Instability of Beams

The deformation instability of beams is characterized mainly by buckling instability, which is generally considered to be a critical failure mechanism that needs to be prevented from occurring in most cases. During the buckling process, there may be multiple modes in the buckling response due to differences in loading forces, temperatures, etc. [152,153,154]. For example, local buckling, global buckling and interactive buckling modes can possibly occur under combing load conditions (as shown in Figure 12a) [155]. Jiao et al. [156,157] studied the buckling process of hollow microstructure beams by numerical simulation, and according to Figure 12b, a significant local buckling can be observed. According to the established dynamic and static theoretical models, the buckling fracture of slender beams constrained by irregular sides can be analyzed. Furthermore, the buckling mode transitions of the beams when subjected to linear and sinusoidal bilateral constraints have been investigated, and the results show that the deformed shape of the beams conforms to constrained modes; both the static and dynamic large deformation models can measure the end shortening, which causes the severe rotation of the neutral axis of beam, as shown in Figure 12c,d. Mhada et al. [158,159] established a multi-scale model of the interaction between global buckling and local buckling by considering the coupling of global and local buckling. It can accurately predict the local buckling region, which provides a theoretical basis for the study of the combined buckling of long-arm beams.

Salem et al. [160] and Yang et al. [161,162] studied the buckling instability of functionally gradient beams. Salem characterized the post-buckling response of beams by establishing a theoretical and numerical simulation model, as shown in Figure 13a. It was found that changing the length of the beam may delay the buckling mode of the next stage by theory and simulation methods, and the feasibility of this phenomenon is verified by experiments. Yang studied the relationship between the free vibration and buckling modes of beams, as shown in Figure 13b. The study revealed that the vibration modes of the functionally gradient composite beams (FG-CB) are nearly identical to their buckling modes. Moreover, under dynamic excitation, FG-CB exhibits a transition from oscillation to elastic instability, making it susceptible to triggering the first instability mode. In addition, Yang also studied the static and dynamic buckling modes of the FG-CB beam, and found that, under different high temperature conditions, the dynamic and static buckling loads of beams are sensitive to the loading position, as shown in Figure 13c. It can be concluded that the number of load limit points is related to temperature and loading position, and that the buckling load decreases with the increment of power exponent, but it increases with the increment of temperature. Wu et al. [163] used an efficient high-order model to study the buckling of functionally gradient sandwich beams, and found that panels reinforced with carbon nanotubes (CNTs) in a uniformly distributed configuration can improve the buckling load and stability of the sandwich beam structure.

## 5. Analytical Methods of Deformation Instability

According to the review of the previous four sections, the research methods of material deformation instability mainly include the establishment of theoretical instability criteria, the use of numerical simulation predictions and experimental verification. Therefore, this section elaborates and summarizes the latest research methods of deformation instability.

### 5.1. Theory Analysis

As we all know, theory serves as the premise for various studies and provides the basis for further investigations. Theoretical analysis of deformation instability research is very important, and is the guarantee for the feasibility of follow-up research results. Therefore, this section mainly elaborates on and summarizes the latest research on deformation instability criteria and instability prediction.

Chawla et al. [164,165] used Timoshenko’s first-order shear deformation beam theory to determine the deflection of I-beam, and calculated the critical buckling load of the flange, web and beam. Based on the failure criterion of strength to calculate the failure load of the beam, they proposed a new I-beam load instability criterion (Table 5, Equation (31)) and the feasibility of the criterion was verified by experiments. In order to study the dynamic instability process of beams, Yang et al. [161] utilized Hamilton’s principle to derive the governing equation of dynamic instability in the thermomechanical plane. They further applied the Bolotin method and solved the equation using the differential quadrature method (DQM). Eyvazian et al. [154] used the nonlinear motion equation to obtain the critical buckling temperature change and critical post-buckling instability. The above research provides a theoretical basis for the analysis of beam instability.

Pozorski et al. [166,167] studied the local instability of a sheet by using the energy method and differential method, and established the instability criterion of the critical wrinkling stress (Table 5, Equation (32)). The sensitivity of the wrinkling stress to changes in material parameters has been verified by parametric analysis, and the feasibility of the theoretical criterion is proven. Wang et al. [168,169] performed a theoretical analysis of the local instability of thin-walled sheet metal during stretching. Under the boundary conditions of force or displacement control, the axisymmetric necking of circular or square hyper-elastic sheets was subjected to equivalent biaxial tension. Furthermore, based on the numerical analysis of strain energy, by comparing the critical force of necking instability and ultimate instability, the moment of necking occurrence can be judged comprehensively. That is, the instability condition for necking does not correspond to Jacobian equal to zero. Sabri et al. [170] studied the wrinkling behavior of composite sheets subjected to in-plane shear deformation. They employed an energy method to establish an in-plane shear model for static analysis, allowing for the determination of potential buckling modes of elastic plate under specified boundary conditions. The theoretical model can predict wrinkle jumping and multiple buckling modes between free boundaries for a certain shear deformation. Huang et al. [171] proposed a theoretical model for calculating the flange forming limit by the energy method, to predict and avoid flange wrinkling of spinning bimetallic clad sheets. According to the theoretical model, the wrinkling forming limit diagram (WLD) of conventional spinning flange was obtained. It was found that the flange wrinkled when the forming angle was larger than the theoretical result. He et al. [172] established a theoretical method by an energy method to determine the instability moment, when the circumferential stress reaches critical wrinkling limit; this method is applied to study the formability of large integral sheets. The stress distribution in the flange area is derived considering the plastic behavior reflected by the plastic modulus, which was caused by the non-axisymmetric shape and the shear stress component. The critical circumferential wrinkling stress of the flange was calculated by the energy method, and a new instability criterion for large sheet metal was obtained (Table 5, Equation (33)).

Li et al. [173,174], based on the radial-axial rolling process of an ultra-large ring with four guide rollers, established a dynamic mechanical model of the combined action of each roll on the ring, and deduced and calculated the bending moment and normal stress. By comparing the normal stress and yield stress, the instability of four guide roller rings was judged and finally the mathematical model of critical instability force was established (Table 5, Equation (34)). Moreover, the plastic instability criterion was validated in terms of critical force, section bending moment, normal stress and plastic instability dangerous ring section; this confirms the reliability of the criterion. Miyajima et al. [175] studied the necking phenomenon of layered metal composites prepared by Accumulative Roll Bonding (ARB), and established a mathematical model for quantitative evaluation and the prediction of instability based on the necking of work-hardened layered metal composites. (Table 5, Equation (35)). The instability index is used to determine the amount of necking, which is used to judge the instability degree of rolled material.

### 5.2. FE Simulation and Experiment

For the study of material deformation instability, theoretical analysis alone is not enough. It must be combined with finite element numerical simulation, experimental research and other comprehensive analysis methods to obtain the best research results quickly and accurately. Therefore, many scholars use finite element simulation, experimental research or a combination of the two methods to analyze the causes of deformation instability, to predict the conditions of deformation instability, or control the behavior of deformation instability.

Malik et al. [176,177,178] used a majority of high-temperature compression experiments to analyze the physical mechanism of deformation instability of magnesium alloys, as shown in Figure 14a. It was found that the shear instability of the magnesium alloy occurred during the compression process under low temperature conditions; as the temperature increased, the sample gradually appeared bulging and cracked along the circumferential direction. Chen et al. [179] combined the 3D machining diagram with FE simulation to simulate the thermal compression process of Cr5 alloy steel, and obtained the distribution and change law of power dissipation and the flow instability domain of metal deformation. When the temperature increases, the instability region decreases. It was also found that the strain had no significant effect on the instability region. Wu et al. [180] established the FLD model based on the instability mechanism of composite sheet forming of the M–K model. The feasibility of FLD was verified by comparing the FE simulation results with a simplified mechanical model. Shuai et al. [181] used a nonlinear FE analysis method to study the compressive strain capacity of an X80 steel corroded pipeline against buckling under the bending moment. It was concluded that, with an increase in the corrosion length of the pipeline, the corrugation waves gradually increased on the bending inner side, thereby augmenting the critical compressive stress and load of the pipeline, as shown in Figure 14b. Nieto-Fuentes et al. [182] investigated the effect of a porous microstructure on the necking formability of a dynamic in-plane tensile plastic sheet. The FE simulation of the dynamic stretching process under different loading conditions showed that the void promotes the localization of plastic deformation, and the position of the void becomes the first position for rapid nucleation, as shown in Figure 14c. Dal et al. [183] investigated the side wall wrinkling of a cylindrical cup during a deep drawing process. The CPB06ex2, BBC2008-8p and BBC2008-16p models were imported into the ABAQUS for numerical simulation; the feasibility of the theoretical model for predicting wrinkling was easily verified.

However, in order to analyze and predict the deformation instability of materials more accurately, most studies take the method of combining finite element simulation and experimental research. Huang et al. [171] analyzed the influence of geometric parameters on flange wrinkling based on the theoretical model of the flange forming limit of spinning thin-walled plates by numerical simulation and experimental research. Through comparative analysis, the FE model can predict wrinkling well and the simulation calculation results are basically consistent with experiments, as shown in Figure 15a. Więckowski et al. [184] optimized the forming process of flange edge wrinkling based on the stamping results of a two-stage mechanical handle of titanium sheet by FE simulation. In stamping processes, the use of an edge holder is imperative to prevent the flange edge from wrinkling. Additionally, the reliability of the simulation optimization results is verified by experiments, as shown in Figure 15b. Chen et al. [185] conducted experiments, simulations and reduced-order calculations on flange wrinkling in the deep drawing process of the AA1100 aluminum alloy. The flange wrinkle height and wrinkle numbers predicted by the FE model and reduced-order model are consistent with experimental results, as shown in Figure 15c. Based on the established sheet deformation instability model, Wang et al. [186] undertook the FE model to compare the prediction results of FLC, maximum stress and strain instability criteria; they found that a new FLC instability criterion can be more accurately predicted and verified for wrinkling. Du et al. [187] took the shear wrinkling experiment of a 304 steel plate as the research object, and used the Buckle-Explicit algorithm in ABAQUS to establish a numerical simulation model of shear wrinkling, as shown in Figure 15d. The accuracy of the simulation algorithm was verified by comparative experiments, and the distribution law of unified critical wrinkling judgment point was analyzed according to numerical simulation results. Li et al. [140] and Lin et al. [188] combined simulation and experimental methods to study the deformation instability of tubes during compression. It was found that dynamic buckling instability is prone to occur during compression, and multiple constraints can cause wrinkling in higher-order buckling modes that consume more energy, as shown in Figure 15e.

## 6. Engineering Applications of Deformation Instability

In engineering applications, the deformation instability of materials may bring potential danger to the service of parts, or directly cause catastrophic accidents; there are also some parts that may take advantage of the deformation instability characteristics of materials to achieve special functions, which can play a key role in special occasions. Therefore, this section summarizes applications in engineering in terms of utilizing and avoiding deformation instability.

In industrial fields such as aerospace, automobile engines, and coastal power stations, axial multilayer compression tubes and variable-diameter tubes play a main role in engine exhaust, fuel transportation and oil transportation because they have the advantages of changing fluid direction and easy installation. The controllable formation of deformation instabilities is essential; however, it is imperative to prevent the occurrence of failures [189]. Variable-diameter tubes are formed through the combined effect of internal pressure and axial force. Under the action of two-way pressure, deformation instability occurs in the tube diameter to a certain extent; at the same time, it is crucial to prevent the occurrence of failures such as rupture and excessive folding [190]. Chu et al. [191,192] and Yuan et al. [193] conducted a detailed study on deforming tubes with variable diameters according to preset paths while preventing the formation of failures. Firstly, the analytical model of the corner wrinkling mechanism was established, and the critical pressure of corner wrinkling was calculated. Based on the method of numerical simulation, it was found that, when the internal pressure used for forming exceeded critical pressure, the corner area failure could be suppressed, but the main failure forms were excessive wrinkling, cracking, etc., as shown in Figure 16a. Haley et al. [126,194] conducted axial compression experiments on Al-6061-T6 round tubes in order to study the forming process of tube compression wrinkling but not cracking, as shown in Figure 16b. It was found that obvious wrinkling appeared in the early stage of compression, and with further compression and bending the surface wrinkling evolved into folds, creases and sharp cracks, which must lead to sectional catastrophic fracture during service. Therefore, in practical engineering applications, deformation instability is sometimes necessary for forming, but it is crucial that it remains predictable and controllable. Moreover, measures must be taken to prevent any reduction in plastic life and subsequent catastrophic failures.

In addition, plastic hinges can realize unidirectional or multidirectional rotation and transmit bending moments in practical engineering applications. The deformation instability of the internal structure of the plastic hinge is used to dissipate energy, increasing the safety reserve and multi-degree of freedom energy release of the actual support structure. Therefore, plastic hinges are widely used in practical engineering, such as in building anti-shock design, automobile anti-collision design and aviation anti-collision design [195,196,197,198,199]. Yuan et al. [200,201] proposed a new performance design method for prefabricated beam-column joints and artificial controllable plastic hinges (ACPH). This design can concentrate the building structure deformation caused by earthquakes on ACPH while preventing damage to the concrete components. ACPH includes a connection system and two energy dissipation systems, and the three-way beam-column joints are staggered according to normal concrete beam-column joints, as shown in Figure 17a. The yield mode of ACPH is the yield of energy dissipation system, and the energy dissipation plate will have deformation instability under the load (as shown in Figure 17b). This design is intended to dissipate the energy of external forces for concrete members and connection systems, enabling the prevention of component damage while also facilitating the rapid recovery of ACPH’s post-earthquake function. However, in order to better improve seismic performance, the design length of the reinforced concrete plastic hinge is strictly limited, so it is necessary to accurately predict the deformation instability region of the plastic hinge [202,203]. Figure 17c shows the internal structure deformation of the vehicle energy-absorbing box [204]; the deformation type is a plastic hinge deformation of half-height. In the compression process, the folding deformation of the plastic hinge is the main type, and can absorb at least 10% of the vehicle kinetic energy during the collision process. This mechanism can effectively protect the interior structure of the vehicle. Zheng et al. [205,206] studied the energy absorption performance of automobile front-end tubes; it was found that when the number of plastic hinges is three, they can absorb energy better and play the role of automobile collision avoidance. Multiple plastic hinges are usually installed in the aircraft cabin to protect the interior safety of the cabin: energy absorbing seats, landing gear and subfloor structures on aircraft are typical applications of plastic hinge deformation and energy absorption [207,208,209,210].

However, in the structural parts and collision parts of aerospace and automotive industries, most of the high-strength and light-weight parts are in the forming process of stamping, spinning, rolling, forging, etc., and deformation instability must be avoided [211,212]. Li et al. [213,214] took advantage of the non-uniform characteristics of material deformation to realize the ring parts forming with different radii in the in-plane roll forming. This forming method has the advantages of small forming force, less material waste, and good flexibility [215]. However, if the forming conditions are not precise controlled, various instability modes such as in-plane wrinkling, external wrinkling, twist instability, and cross-section instability will appear (as shown in Figure 18a); these defects lead to scrapped parts. Therefore, it is necessary to control the occurrence of wrinkling instability. Chen et al. [216,217] analyzed the buckling and wrinkling of curved shell sheet metal hydroforming in the presence of an anti-bulging effect. Utilizing the energy method and incorporating the appropriate “anti-bulging effect”, the authors proposed a theoretical model of critical wrinkling stress. By combining the critical wrinkling stress with circumferential stress, the critical loading path of hydraulic pressure to control wrinkling was obtained. In addition, through experimental research, it was concluded that when using sheet metal hydroforming technology, as long as there is an appropriate “anti-expansion effect”, the wrinkling of unsupported sheet metal can be eliminated effectively, as shown in Figure 18b(3–4). However, if the hydraulic pressure is too large, cracks will appear in the raised part, and the flange part will show wrinkling (as shown in Figure 18b(5–6)) [218]. References [123,219,220,221] studied the deformation instability behavior of aircraft fuselage and automotive body parts during stamping. Lightweight alloys are predominantly utilized in the manufacture of fuselage skins, upper and lower wing skins, wing girders, wing ribs, car roof rails, etc., among other components that demand exceptionally high levels of forming precision. High-strength steel (HSS) and ultra-high strength steel (UHSS) are widely used in the lightweight design of automobiles. The flange and flange edge are extremely easily wrinkled during forming process (as shown in Figure 18c,d), and flange wrinkling can be reduced by varying the pressure and blank-holder clearance [123,219,220]. Atxaga [221] studied the wrinkling instability of an AA2198 aluminum alloy in different stamping processes. It was found that, under cold forming conditions, the part side was prone to wrinkling, wall thickness thinning or even cracking, and wrinkling was significantly improved under hot forming conditions (as shown in Figure 18e).

In conclusion, the deformation instability of materials is extremely critical in engineering applications. According to the specific forming conditions and forming requirements, catastrophic failure must be avoided and deformation instability should be used reasonably for process forming.

## 7. Conclusions and Outlook

The study of deformation instability is very important in engineering applications, and it has always been the focus of the stability of material forming. Previous research has achieved relatively perfect results in terms of theoretical analysis solutions and other research methods on metal deformation instability. This paper comprehensively analyzes, summarizes and reviews the latest theoretical methods, discoveries and achievements in the study of the deformation instability of metal materials, helping more researchers to quickly extract the latest research results in this field, and proposing more innovative research content. This work has a high reference significance. Additionally, the conclusions are as follows: (1) Based on the classical instability theory, more modified deformation instability criteria have been proposed and deduced. They can be used to accurately predict and control deformation instability, and to achieve the purpose of improving the precision of metal forming. (2) Inherent characteristics of materials or new properties caused by changes in forming conditions all produce different deformation instability characteristics. Notably, instability characteristics caused by the latter are typically more complex. (3) The deformation instability characteristics of materials are closely related to their macro-micro structures, manifesting primarily in the form of wrinkling, buckling, etc. (4) The main methods of deformation instability research include theoretical analysis and FE simulation and experiment, and most studies combine two or three methods to obtain feasible conclusions to improve the forming accuracy. (5) Deformation instability of materials plays a crucial role in engineering applications. Depending on the specific forming conditions and forming requirements, it is essential to avoid catastrophic failure and utilize deformation instability in a reasonable manner for process forming.

However, there are still many key problems in the study of deformation instability. In practical engineering applications, there are many factors affecting deformation instability, and the forming conditions and forming environments of materials are very complex. In previous studies, some conditions have been idealized and the modal technology for deformation instability analysis was not accurate enough; there is also a lack of detailed research on different proportion superposition analyses of various deformation instability modes. The related research is all about finding the critical load to predict instability, but the proportion of the impact of load on the different instability states is unknown. When the applied external conditions change, it can be difficult to predict the location of instability using the existing analysis method. Similarly, if, in the complex environment of engineering accidents, it is known that the parts have several instability states, it can be difficult to know how to quickly obtain the critical load at that time, which requires new theoretical analysis and analytical solution. With the rapid development of informatization and industrialization, new materials are continuously being developed and used. Whether the previous related research is applicable to the deformation instability analysis of these new materials still needs to be debated and verified.

Therefore, research on the deformation instability of metal materials should mainly focus on the complex forming conditions of practical engineering applications and the forming analysis of new materials. Additionally, using reverse thinking to solve practical engineering problems, proposing universal instability criteria, broadening the application of instability criteria, and conducting correlation analysis with deformation damage, instability fracture, and service life changes caused by deformation, are the inevitable trends of future development.

## Figures and Tables

**Figure 1 materials-16-02667-f001:**
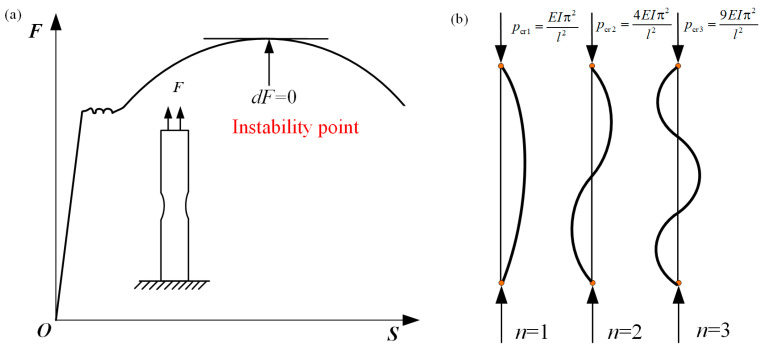
Types of plastic instability: (**a**) tensile instability; (**b**) compression instability [24].

**Figure 2 materials-16-02667-f002:**
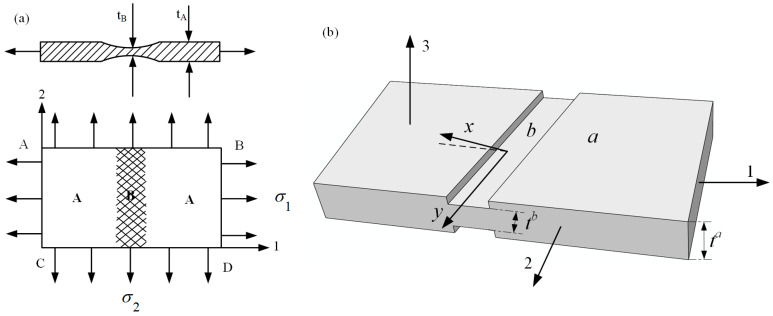
M–K model: (**a**) classical model [30]; (**b**) modified model [34].

**Figure 3 materials-16-02667-f003:**
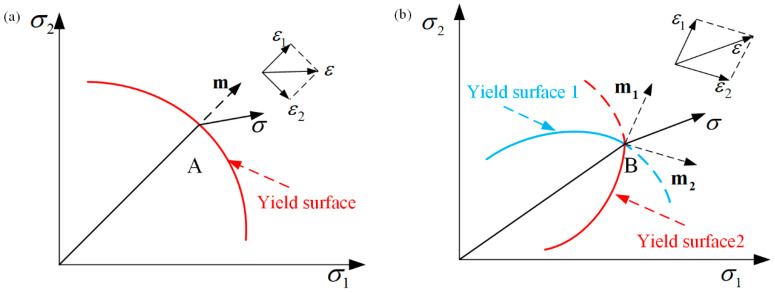
Schematic diagram of Storen and Rice model [48]: (**a**) plastic flow theory with ideal yield surface; (**b**) plastic flow theory with singular point.

**Figure 4 materials-16-02667-f004:**
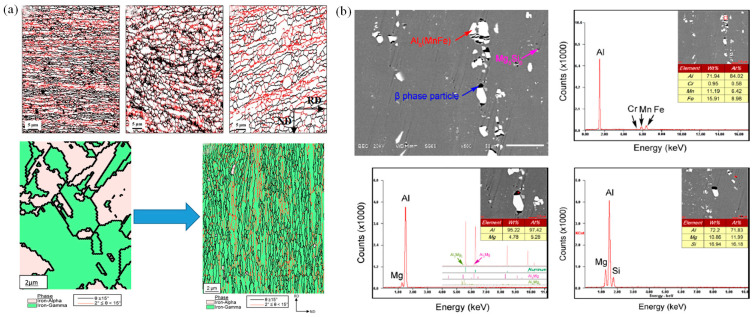
The superplastic deformation instability behavior of an aluminum alloy: (**a**) grain boundary slip leads to superplastic deformation instability [75,76]; (**b**) fine precipitates dispersed during the superplastic deformation process [77].

**Figure 5 materials-16-02667-f005:**
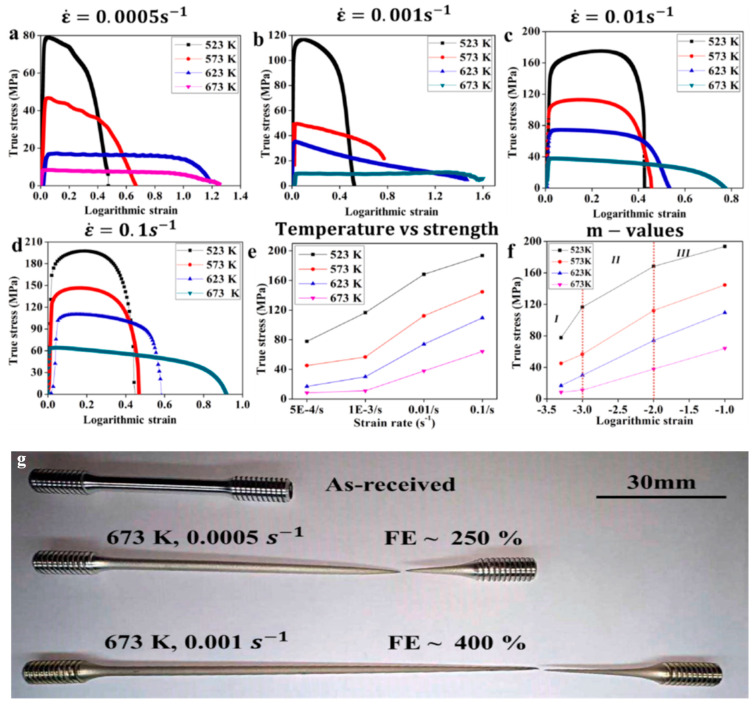
Superplastic deformation behavior of the Mg alloy: (**a**–**f**) are the real stress–strain curves at different temperatures and different deformation rates; (**g**) is the superplastic behavior of the actual tensile specimen [78].

**Figure 6 materials-16-02667-f006:**
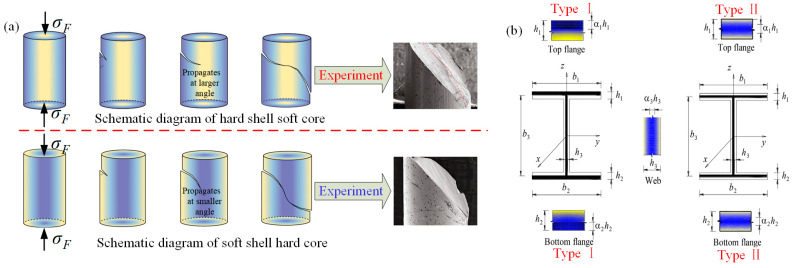
Deformation instability mechanism of functionally gradient metal materials: (**a**) shear instability and actual fracture morphology of hard-shelled soft-core and soft-shelled hard-core materials [88]; (**b**) schematic diagram of two types of sandwich I-beams [89].

**Figure 7 materials-16-02667-f007:**
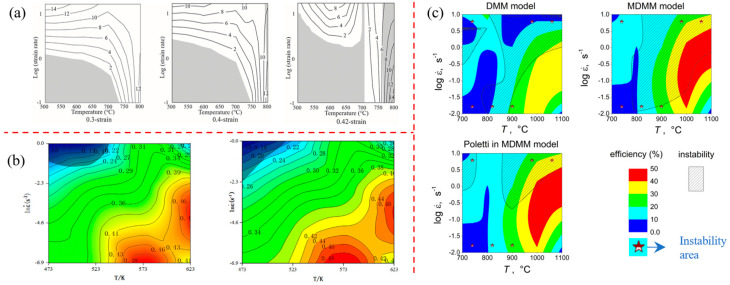
Hot processing diagrams of different materials during compression: (**a**) the gray area is the deformation instability area of the Ti-6Al-4V titanium alloy [94]; (**b**) the dark blue area is the deformation instability area of the dual-phase Mg-9Li-3Al alloy [96]; (**c**) ★ indicates the deformation instability point of stabilized micro-alloyed steel [98].

**Figure 8 materials-16-02667-f008:**
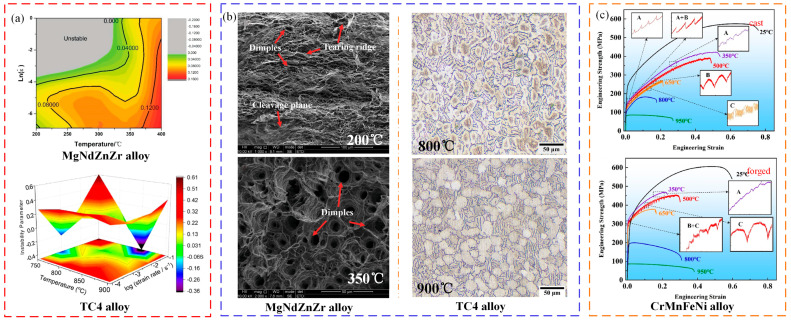
Deformation instability characteristics of different materials under hot tensile conditions: (**a**) hot processing diagrams of MgNdZnZr alloy and TC4 alloy; (**b**) fracture morphology of MgNdZnZr alloy and TC4 alloy at different temperatures [65,101]; (**c**) engineering stress–strain curves of CrMnFeNi high-entropy alloys, with as-cast and forged samples [103].

**Figure 9 materials-16-02667-f009:**
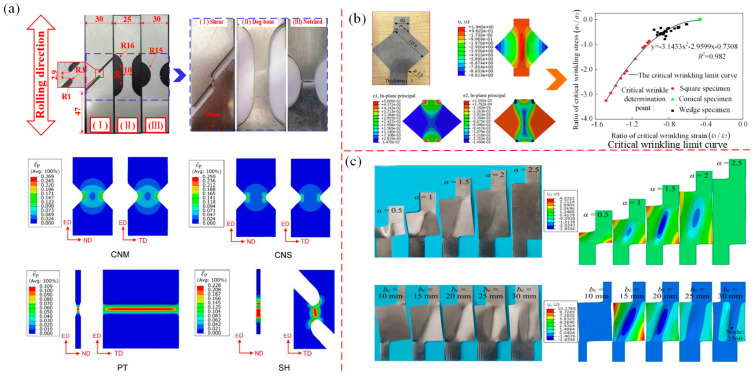
Deformation instability characteristics of sheet metal stretching: (**a**) local instability of a notched sheet metal specimen [117,118]; (**b**) wrinkling of the thin plate when stretched symmetrically [115,119]; (**c**) wrinkling of the thin plate when stretched asymmetrically [120].

**Figure 10 materials-16-02667-f010:**
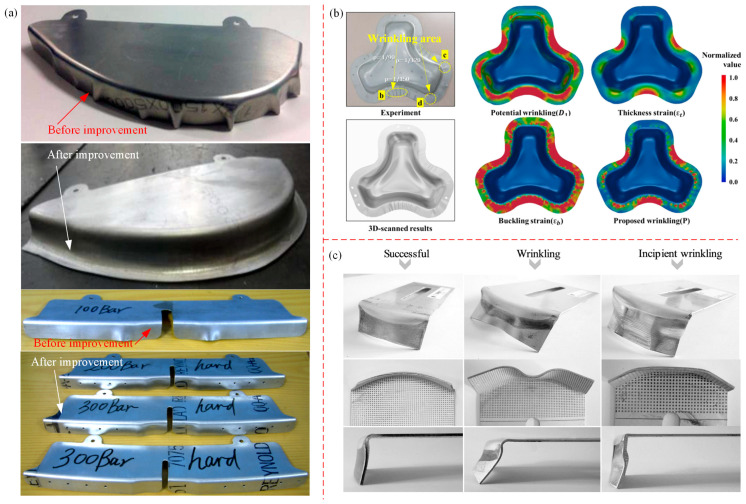
Wrinkling behavior of sheet metal stamping: (**a**) sheet metal stamping of a high curvature large flange [121,122]; (**b**) wrinkling of GPa-grade steel, comparison analysis of representative wrinkle area (wrinkling area b, c and d) with the simulation results [123]; (**c**) failure mode of a AA2024-T3 thin plate [124].

**Figure 11 materials-16-02667-f011:**
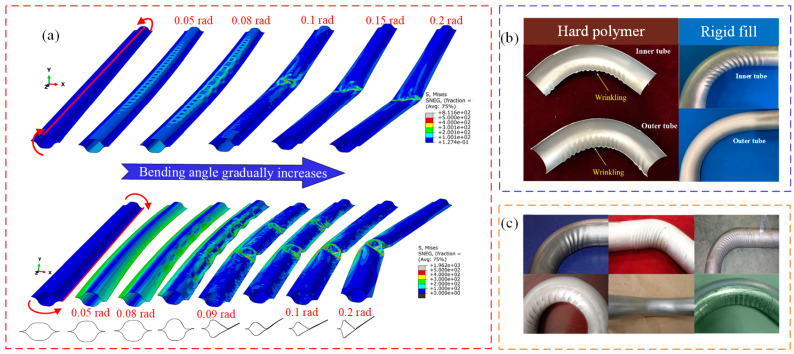
Wrinkling instability of the tube: (**a**) wrinkling of the composite thin-walled lenticular tube [132,133]; (**b**) the wrinkling comparison of the inner and outer tubes under different filling conditions [137,141]; (**c**) the wrinkling position of different tubes [140].

**Figure 12 materials-16-02667-f012:**
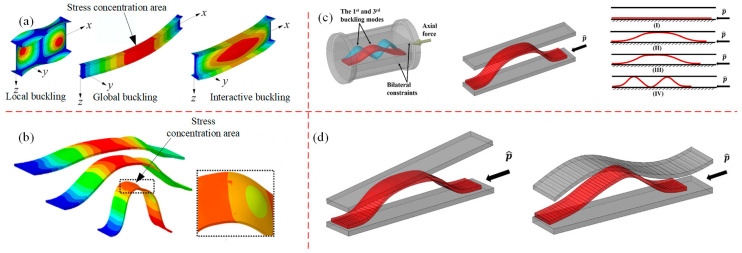
Modes of beam buckling: (**a**) local buckling, global buckling and interactive buckling [155]; (**b**) local buckling of a hollow microstructure beam [156]; (**c**) buckling schematic diagram of a slender beam with irregular bilateral constraints; (**d**) buckling modes of beams under linear and sinusoidal two-sided constraints [157].

**Figure 13 materials-16-02667-f013:**
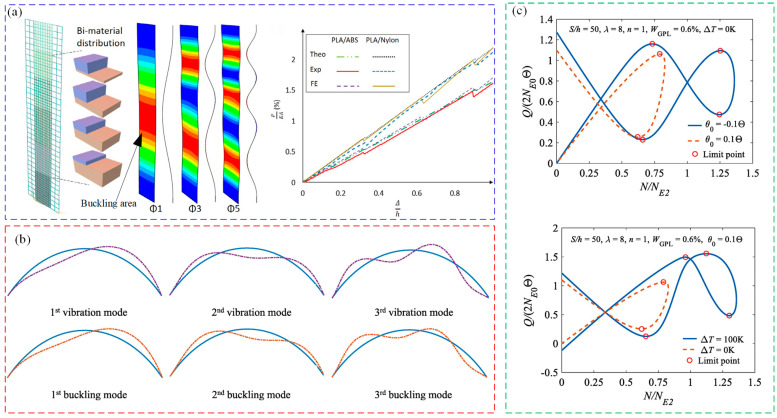
Buckling instability of functionally gradient beams: (**a**) numerical simulation, modal and experimental verification results of laminated functionally gradient materials (FGM) beams [160]; (**b**) relationship between free vibration and buckling modes of functionally gradient composite beams (FG-CB) [161]; (**c**) relationship of FG-CB load limit point number and loading position [162].

**Figure 14 materials-16-02667-f014:**
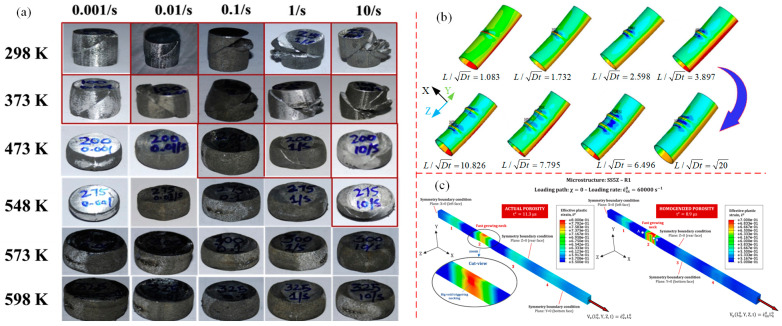
Only one research method using experiment or simulation: (**a**) hot compression experiment of magnesium alloy [176]; (**b**) wrinkling simulation of X80 steel tube [181]; (**c**) dynamic tensile simulation of porous microstructure [182].

**Figure 15 materials-16-02667-f015:**
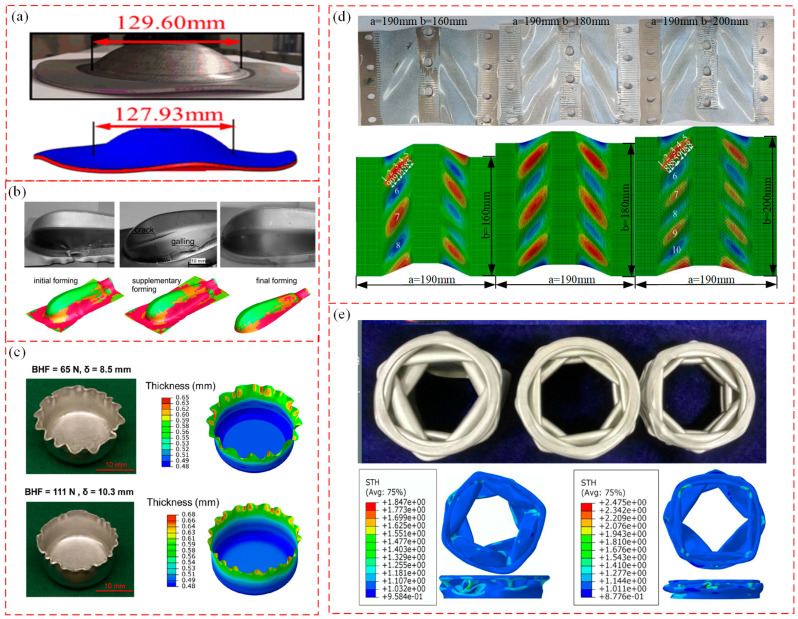
Research results of deformation instability by finite element simulation and experimental methods: (**a**) flange forming of the spinning thin-walled plate [171]; (**b**) stamping forming of a titanium alloy mechanical handle [184]; (**c**) deep drawing of the AA1100 aluminum alloy [185]; (**d**) 304 steel plate shear wrinkling test [187]; (**e**) various buckling modes of tube compression instability [140].

**Figure 16 materials-16-02667-f016:**
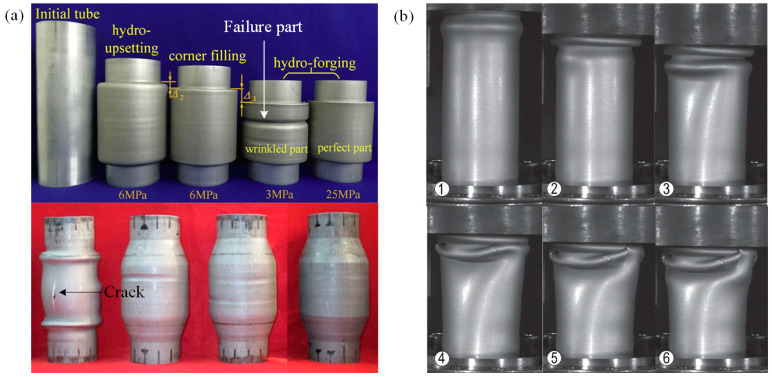
Forming by deformation instability: (**a**) forming of variable-diameter tube [192,193]; (**b**) tube forming with axial compression, and analyzing the degree of deformation instability in the six-step (①–⑥) compression process [194].

**Figure 17 materials-16-02667-f017:**
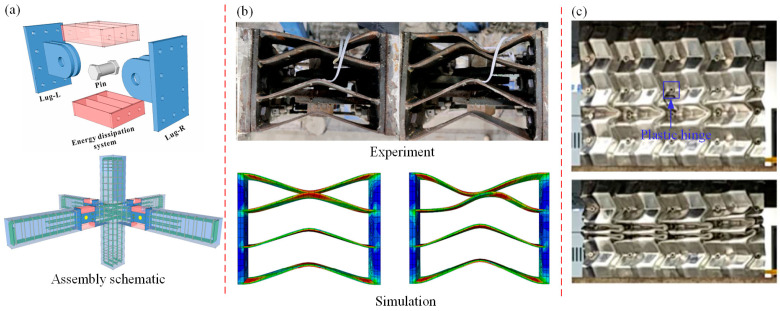
Application of plastic hinges in engineering: (**a**) schematic diagram of ACPH; (**b**) plastic hinge deformation of ACPH energy dissipation board [200,201]; (**c**) plastic hinge deformation of automobile energy-absorbing box [204].

**Figure 18 materials-16-02667-f018:**
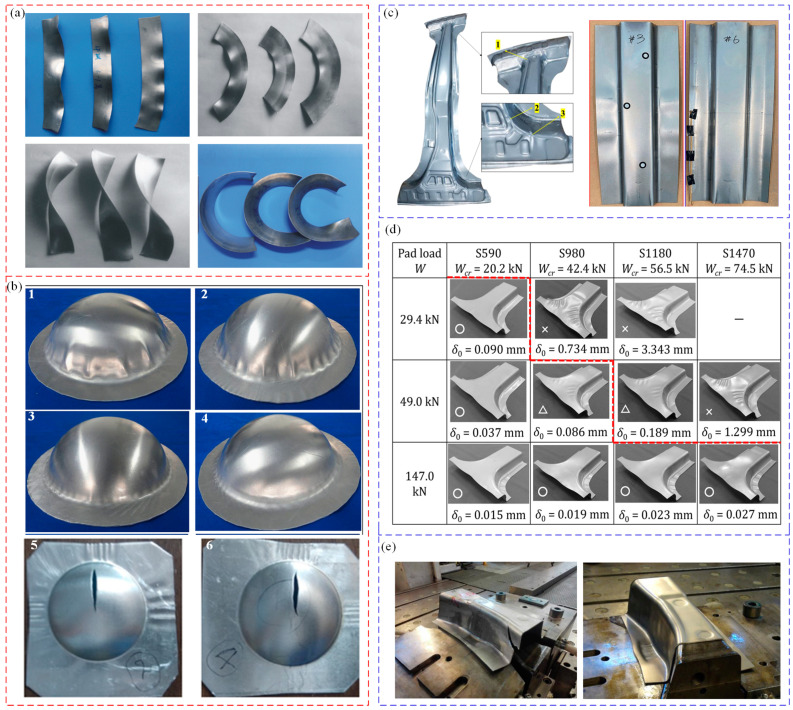
Engineering applications of avoiding deformation instability: (**a**) in-plane bending of plates [213]; (**b**) hydroforming of sheet metal [216,218]; (**c**) stamping of ultra-high strength steel [123,219]; (**d**) stamping of high tensile strength steel [220]; (**e**) stamping of AA2198 aluminum alloy [221].

**Table 1 materials-16-02667-t001:** Summary of research results on modified M–K model in recent years.

Authors	Materials	Modified M–K Model	Representative Figures
Wang et al. [35,36]	6061 Aluminum Alloy	${\left(\sum \beta^{A}d\varepsilon_{1}^{A}\right)}^{n^{A}}\exp\left(\varepsilon_{1}^{B}-\varepsilon_{1}^{A}\right)\varphi^{B}=f_{0}{\left(\sum \beta^{B}d\varepsilon_{1}^{B}\right)}^{n^{B}}\varphi^{A}$ where $\varphi =\sigma /\sigma_{1}$, $\beta =d\varepsilon /d\varepsilon_{1}$, *n* is real-time strain hardening exponents, and *A* and *B* are zone-*A* and zone-*B*, respectively.	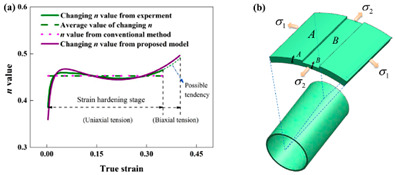
Yu et al. [37,38]	AA5182O Sheet	$\psi^{*}=\left\{ \begin{array}{ll} arctan\sqrt{-\rho }, & -0.5\leq \rho \leq 0\\ 0, & 0<\rho \leq r_{{}_{0}}/r_{max}\\ \frac{\rho \cdot r_{max}-r_{{}_{0}}}{r_{max}-r_{{}_{0}}}\theta_{r-max}, & r_{{}_{0}}/r_{max}<\rho \leq 1 \end{array}\right.$ where $\psi^{*}$ is critical groove angle, ρ is strain routes, $r_{{}_{0}}/r_{max}$ is degree of anisotropy, and $\theta_{r-max}$ is maximum *r*-value.	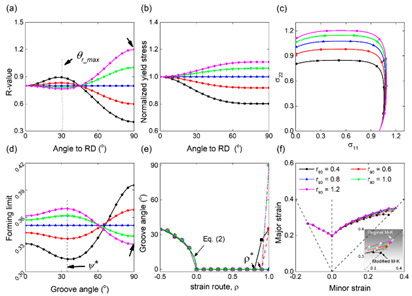
Hyuk et al. [39]	Ferritic Stainless Steel (FSS) Sheets	$\sigma_{XX}^{B}=\sigma_{XX}^{A}\left(\frac{t^{A}}{t^{B}}\right)\left(\frac{t_{0}^{A}}{t_{0}^{A}-2R}\right)$ where *t* is thick, *t*_0_ is initial thick, *R* is surface roughness, and $\sigma_{XX}^{A}$ and $\sigma_{XX}^{B}$ are the stresses of REV-A and B, respectively.	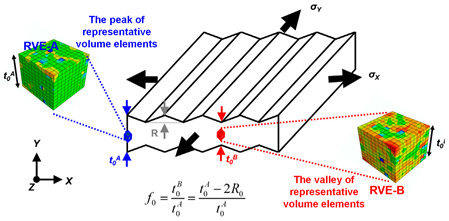
Wang et al. [40,41,42,43]	Al-Mg-Li Alloy Sheet	$f_{0}=t_{0}^{b}/t_{0}^{a}$ $f=f_{0}\exp\left(\varepsilon_{3}^{b}-\varepsilon_{3}^{a}\right)$ where *f*_0_ is imperfection coefficient and $t_{0}^{a}$ and $t_{0}^{b}$ are initial thick of *a* and *b*, respectively.	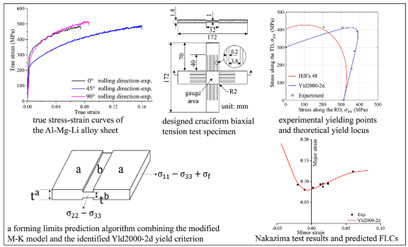
Li et al. [44]	Aluminum Alloy	$e=\left\{ \begin{array}{ll} \sigma_{1}^{A}t^{A}-\sigma_{1}^{B}t^{B} & \mathrm{Classical}\;\text{M-K}\;\mathrm{model}\\ {\left(\varepsilon_{i}^{A}+d\varepsilon_{i}^{A}\right)}^{n}t_{0}^{A}\exp\left[-\left(\varepsilon_{i}^{A}+d\varepsilon_{i}^{A}\right)\right] & \\ -{\left(\varepsilon_{i}^{B}+d\varepsilon_{i}^{B}\right)}^{n}t_{0}^{B}\exp\left[-\left(\varepsilon_{i}^{B}+d\varepsilon_{i}^{B}\right)\right] & \mathrm{Modified}\;\text{M-K}\;\mathrm{model} \end{array}\right.$	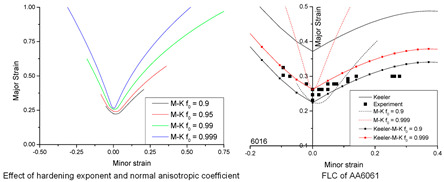
Hu et al. [45,46]	AA5754 Aluminm Alloy	$\begin{array}{ll} \sigma_{nn} & =\left(\sigma_{1}-\sigma_{3}\right)\cos^{2}\theta +\left(\sigma_{2}-\sigma_{3}\right)\sin^{2}\theta -2\sigma_{12}\sin\theta \cos\theta \\ & =\left(\cos^{2}\theta +\alpha \sin^{2}\theta -2\delta \sin\theta \cos\theta \right)\left(\sigma_{1}-\sigma_{3}\right) \end{array}$ $f=f_{0}\exp\left\{\left[\varepsilon_{1}^{a}+\varepsilon_{2}^{a}+\Delta \varepsilon_{1}^{a}\left(1+\beta_{a}\right)\right]-\left[\varepsilon_{1}^{b}+\varepsilon_{2}^{b}+\Delta \varepsilon_{1}^{b}\left(1+\beta_{b}\right)\right]\right\}$ where $\alpha =\left(\sigma_{2}-\sigma_{3}\right)/\left(\sigma_{1}-\sigma_{3}\right)$; $\delta =\sigma_{12}/\left(\sigma_{1}-\sigma_{3}\right)$.	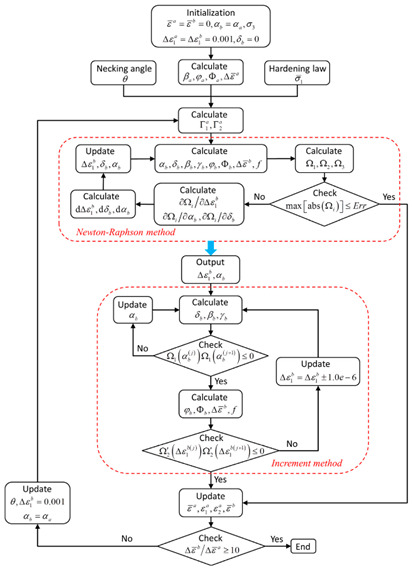
He et al. [47]	AA6061-F Tube	$f_{0\theta }=\frac{{\bar{t}}_{B0}}{{\bar{t}}_{A0}}=\frac{t_{\min}}{R_{1}-\sqrt{\left(R_{2}^{2}-\sin^{2}\theta \cdot \Delta d^{2}\right)}-\Delta d\cdot \cos\theta }$ $f_{0\theta }=\frac{{\bar{t}}_{B0}}{{\bar{t}}_{A0}}=\frac{t_{\min}}{R_{1}-R_{2}-\Delta d\cdot \cos\theta }=\frac{\bar{t}-\Delta d}{\bar{t}-\Delta d\cdot \cos\theta }$ where *t*_min_ is minimum thickness on the tube, θ is angle, ${\bar{t}}_{A0}$ and ${\bar{t}}_{B0}$ are the thicknesses of zone-A and zone-B, Δ*d* is the eccentric distance of extrusion mandrel, and *R*_1_ and *R*_2_ are the radius of outer and inner profiles of the tube, respectively.	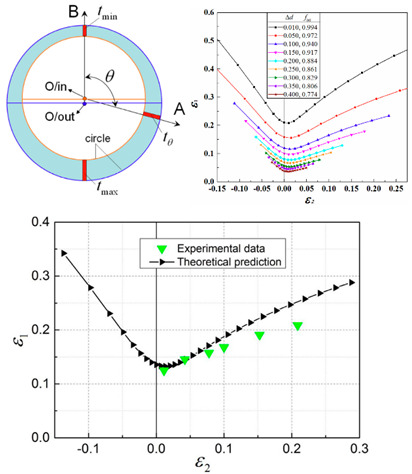

**Table 2 materials-16-02667-t002:** A review of recent research on the superplastic deformation instability of alloy materials.

Author	Year	Material	Main Points
Demirel et al. [80]	2023	Ti6Al4V	For high-temperature superplastic formation of Ti alloys, the main causes of deformation instability are grain boundary slip (GBS) and creep mechanisms.
Bobruk et al. [81]	2023	2021Al	For ultrafine grained (UFG) Al alloys, according to the analysis of strain rate sensitivity, they showed stable superplastic behavior at the test temperature of 240~270 °C.
Myshlyaev et al. [82]	2023	Al-Mg-Li	The important role of intra-grain slip during superplastic flow was demonstrated through experimental analysis of strain hardening, the formation of typical deformation textures, and the increase of dislocation density within grains. Superplastic materials exhibited pronounced porosity near the instability point.
Mochugovskiy et al. [83]	2023	Al-Mg-Si-Cu	When the strain rate was low, the residual cavitation after superplastic forming was relatively large; the impurity particles inside the grains also caused the surrounding cavities to increase, which would easily lead to superplastic deformation instability.

**Table 3 materials-16-02667-t003:** Summary of the latest research on deformation instability in hot forming.

Authors	Year	Material	The Conditions of Deformation Instability	
			Temperature/°C	Strain Rate/s^−1^
Shabani et al. [104]	2023	FeCrCuMnNi	750~850	0.1, 0.01, 0.001
Singh et al. [105]	2023	EN30B Steel	1000~1150	0.1~0.8
Jeong et al. [106]	2023	AlSi4340 Steel	1000~1100	0.1, 0.2, 0.9, 1.0
Azizi et al. [107]	2023	AlSiAA4032	427~527	0.01~0.1
Yang et al. [108]	2023	Al4.6Mg0.2Sr	300~400, 400~450	0.018~1, 0.018~0.1
Lin et al. [109]	2022	Ti47.5Al2.5V1.0Cr0.2Zr	1050~1140, 1180~1200	0.006~1
Yang et al. [110]	2022	215AlLi	390~520	0.1~10
Qiao et al. [111]	2022	Fe2.5Ni2.5CrAl	1020~1100	0.01~1
Ghosh et al. [112]	2022	Ti14Cr	850~950	0.01
Yi et al. [113]	2022	Al0.5Mg0.4Si0.1Cu	350~500	0.316~10

**Table 4 materials-16-02667-t004:** Summary of deformation instability during bending tube process.

Authors	Materials	Types of Forming	Conditions of Instability
Yu et al. [143]	ST12	Push bending	Gap between punch and U/O die, and excessive stock at the end of elbow causing wrinkling.
Tao et al. [144]	5A02 Al Alloy	Push bending	Due to the tangential tensile stress concentration at the front end of the tube, the smaller the relative bending radius, the easier it is to have instability.
Xiao et al. [145]	5A02 Al Alloy	Push bending	The stress distribution on the compression side is greater than the tension side, indicating that inner side of the tube is more prone to instability.
Österreicher et al. [146]	AA2024	Three-roll-push bending	Only solution-annealed material leads to a wrinkle-free bend.
Cheng et al. [147]	AA6061-T6	Free bending	When *t*_0_ < 0.8 mm, plastic instability and wrinkling occurred in the inner flange, and the smaller the wall thickness, the more obvious the wrinkling.
Wang, Hu and Cheng et al. [148,149,150]	Stainless Steel SS304	Free bending	The smaller the distance between the center point of the bending die and the front end of guide, the easier it is for the tube to wrinkle.
Yang et al. [151]	SS304	Free bending	The inner side of the rectangular tube is subjected to uneven compressive stress, which makes the material flow unevenly, resulting in increased wall thickness on the inner side of the tube, and wrinkled instability.

**Table 5 materials-16-02667-t005:** Summary of instability criteria analytical equations.

Authors	Equation		Explanation
Chawla et al. [164]	${\left(\frac{\sigma_{lb}}{S_{lb}}\right)}^{2}+{\left(\frac{\sigma_{tc}}{S_{tc}}\right)}^{2}+{\left(\frac{\tau_{xz}}{S_{xz}}\right)}^{2}\leq 1$	(31)	$\sigma_{lb}$$,\sigma_{tc}$$\;\mathrm{and}\;\tau_{xz}$ are longitudinal compressive bending stress, transverse compressive stress and shear stress, respectively; $S_{lb}$, $S_{tc}$ and $S_{xz}$ are bending compressive strength, transverse compressive strength and shear strength, respectively.
Pozorski et al. [166]	$\begin{array}{l} \sigma_{w}=\sqrt[{3}]{\frac{3}{4}}\cdot \sqrt[{3}]{E_{C}G_{C}E_{F}}≅0.909\cdot \sqrt[{3}]{E_{C}G_{C}E_{F}}\\ \sigma_{w}=\frac{3}{2\cdot \sqrt[{3}]{6}}\sqrt[{3}]{E_{C}G_{C}E_{F}}≅0.825\cdot \sqrt[{3}]{E_{C}G_{C}E_{F}}\\ \sigma_{w}=\sqrt[{3}]{\frac{9}{2\left(1+v_{c}\right)\cdot {\left(3-v_{c}\right)}^{2}}}\cdot \sqrt[{3}]{E_{C}G_{C}E_{F}}=r\cdot \sqrt[{3}]{E_{C}G_{C}E_{F}} \end{array}$	(32)	*E_C_* and *G_C_* are the modulus of elasticity and shear modulus of the isotropic core material; *E_F_* is the modulus of elasticity of the isotropic facing material.
He et al. [172]	$\sigma_{cr}=D\frac{\gamma^{2}\sqrt{2\beta +1}\left[s_{1}\beta^{2}\delta^{3}+s_{1}S\delta^{3}+2s_{2}\zeta \beta^{4}\left(\delta -1\right)/\left(\beta -1\right)\right]}{12\sqrt{k}\sqrt{S}\beta^{2}\delta^{3}\left[\beta -1-\ln\beta +\eta \left(\beta -1\right)\left(\delta -1\right)/\delta \right]}$	(33)	*D* is the plastic modulus; *k* is a coefficient related to the flange width and Poisson’s ratio, *k* = 1.5; γ,β,ζ,η are coefficients related to the geometrical parameters of the material; $s_{1}\;\mathrm{and}\;s_{2}$ are the coefficients representing the increase in the moment of inertia caused by the shift of the neutral surface after stiffening; S=k/[4(2β+1)].
Li et al. [173]	$\sigma_{\max}=\frac{{\left|M\right|}_{\max}}{W_{y}}=\frac{6R_{t}F_{G1}{\left|Q_{k}\left(\alpha_{1},\alpha_{2},\varphi ,k_{g}\right)\right|}_{\max}}{hb^{2}}\leq \sigma_{s}$	(34)	${\left|M\right|}_{\max}$ is the maximum section bending moment; *W_y_* is section modulus in bending; *h* and *b* are the width and height of the section, respectively; *R_t_* is the radius; $Q_{k}\left(\alpha_{1},\alpha_{2},\varphi ,k_{g}\right)$ is section bending moment factor; $F_{G1}$ is guide forces.
	$I=\int_{\varepsilon_{b}}^{\varepsilon_{a}}Bd\varepsilon_{eq}=\int_{\varepsilon_{b}}^{\varepsilon_{a}}\left\{\left(\frac{d\sigma^{h}}{d\varepsilon_{eq}}-\frac{d\sigma^{s}}{d\varepsilon_{eq}}\right)-\left(\sigma^{h}-\sigma^{s}\right)\right\}d\varepsilon_{eq}$	(35)	$\varepsilon_{eq}$ is equivalent strain; h and s are hard and soft floor, respectively; *B* < 0 indicates necking progression, larger absolute values.

## Data Availability

Not applicable.

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
