# Peer review of "Theory, Method and Practice of Metal Deformation Instability: A Review"

_materials, 2023, doi:10.3390/ma16072667_

Round 1
Reviewer 1 Report
Remarks:
1. Introduction
1). In the introduction, (line 61) the authors write: "So, I it is further summarized the latest literature on deformation instability in practical engineering applications, proposed the unresolved problems and future research directions of deformation instability”.
The article was prepared by 5 authors. Therefore, in the sentence, the word "I" should be replaced with the word "We".
2. Definition of deformation instability
2). (line 70) The authors write «In the 18th century, based on the problem of elastic compression rods, Euler proposed the definition of compression rod instability firstly… [3]».
Однако, статья [3], на которую ссылаются авторы, написана в 21th century (line 926).
3). (lines 97-99) The authors write «Tensile instability includes dispersion instability and concentration instability… a metastable flow occurs in a relatively wide area, which is called dispersion instability».
(lines 99-101) The authors write «However, when the tensile necking expands to a certain extent, the unstable flow will be confined to a narrow area, which is called local instability or concentration instability».
(lines 206-209) The authors write «Geometric instability is the process of materials non-uniform deformation, that is, when the uniform strain is greater than strain hardening exponent, the strain hardening effect cannot offset the increase in stress caused by contraction of cross-section, geometric instability occurs at this time».
The authors use a large number of types (names) of instability when considering, which makes it difficult to understand the process of deformation instability.
7. Conclusions and outlook
4). (line 884) The authors draw conclusions about the importance of studying deformation instability in engineering applications.
However, since the authors formulated the topic of their article "Theory, Method and Practice of Metal Deformation Instability: A review", I would like to summarize how from the 18th to the 21st centuries the understanding in theory, science and practice has changed to the processes of deformation instability.
Author Response
Thank you very much for your kind and carefully review. The detailed content of reponse is in the PDF, please see the attachment.

Reviewer 2 Report
The author has made a significant effort on a review -Theory, Method and Practice of Metal Deformation Instability. however a few corrections needs to be incorporated in order to enhance the quality of manuscript that are given below
1. The title of the paper was in written generalized mode. point out in which the review was focused
2. Abstract needs to be modified as problem identification, intervention, comparison and outcome
3. The authors has given lot of datas. Sine it is an research paper but it was inferiorly cooked like thesis. Instructed to prepare like manuscript. It mandatory to categorize what are things needful , remove unwanted and known datas
4. Novelty is more important for the manuscript. It must be included in the manuscript
5. Line no 194 (In 1984, Semiatin, et.al), line no 200 (In the 20th century)- remove all these stories. Since it was known to all
6. Since it was a review papers, only relavant datas needs to be discussed. So that it makes readers easy to understand. Lot of cooked input datas goes out of syllabus. Optimize it
So the authors are request to look out the above corrections and submit. Futher it may be recommended for publications
Author Response

(The authors gave the same response as above.)

Reviewer 3 Report
Manuscript ID: materials-2261961 entitled “Theory, Method and Practice of Metal Deformation Instability: A review” for journal of “Materials” has been reviewed.
This review article is comprehensive, logically organized and contains valuable information. However, there are few things need to be corrected and included in the manuscript for better understanding of carried research work to the readers.
+1- The important points of the study should be given in the Abstract.
+2- The introduction section should be enriched.
+3- More references (different studies) should be added to the introduction. (recent studies, 2020-2022).
+4- The resolution of Figure 2, 3, 4, 5, 6, 7, 8 and 9 should be increased. (and magnify) (and Table 3)
+5- … Chawla et.al[143, 144] used Timoshenko's first-order shear deformation beam theory to determine the deflection of I-beam, and calculated the critical buckling load of the flange, web and beam; based on the failure criterion of strength to calculate the failure load of the beam, a new I-beam load instability criterion is proposed (Table 5, Equation 31), and the feasibility of criterion is verified by experiments. In order to study the dy namic instability process of beam, Yang et.al[140] derived the governing equation of the dynamic instability in thermomechanical plane by using the Hamilton’s principle, and combined with the Bolotin method, the differential quadrature method (DQM) is used to solve it. Eyvazian et.al [133] used the nonlinear motion equation to obtain the critical buck ling temperature change and critical post-buckling instability. The above research pro- vides a theoretical basis for the analysis of beam instability. Pozorski et.al[145, 146] used the energy and differential method to study the local instability of sheet, and respectively established the instability criterion of the critical wrinkling stress (Table 5, Equation 32). The sensitivity of wrinkling stress to changes in material parameters is verified by parametric analysis, proving the feasibility of the theo retical criterion. Huang et.al [147] proposed a theoretical model for calculating flange forming limit by the energy method, to predict and avoid flange wrinkling of spinning bimetallic clad sheets. According to the theoretical model, wrinkling forming limit diagram (WLD) of conventional spinning flange is obtained, and the wrinkling is predicted and analyzed. Sabri et.al [148] studied the wrinkling behavior of composite sheets sub jected to in-plane shear deformation. An in-plane shear model is established for static analysis, to determine the potential buckling modes of elastic plate under specified boundary conditions. Wang et.al[149, 150] performed a theoretical analysis of the local instability of thin-walled sheet metal during stretching. Under the boundary conditions of force or displacement control, axisymmetric necking of circular or square hyper-elastic sheets subjected to equivalent biaxial tension. Meanwhile, based on the numerical analysis of strain energy, comparing the critical force of necking instability and ultimate instability, the moment of necking occurrence can be judged comprehensively. He et.al[151] estab- lished a theoretical analysis method to determine the moment when the circumferential stress reaches critical wrinkling limit, in order to study the formability of large integral sheets. Considering the plastic behavior reflected by plastic modulus caused by the non-axisymmetric shape and shear stress component, the stress distribution in flange area is deduced; The critical circumferential wrinkling stress of flange is calculated by energy method, and a new instability criterion for large sheet metal is obtained… This section should be re-examined and rewritten.
+6- Conclusions section should be enriched a little more.
+7- More literature studies should be added to the introduction and other sections (DOIs given below).
DOI-1 https://doi.org/10.26701/ems.1109917 (different studies)
DOI-2 https://doi.org/10.36222/ejt.1086422 (different studies)
---------------------------------------------
*The article will be ready for publication after the specified revisions are made.
**After revision, I would like to review the article again.
------------------------------------------------------
Congratulations to the authors.
I wish the authors success in their future academic studies.
Kind regards.
Author Response

(The authors gave the same response as above.)

Round 2
Reviewer 2 Report
As suggested by reviewer all comments was verified and it may be recommended for publication in the present form
Author Response
Thank you very much for your carefully review.
Kind regards.
Reviewer 3 Report
Manuscript ID: materials-2261961 entitled “Theory, Method and Practice of Metal Deformation Instability: A review” for journal of “Materials” has been reviewed.
The authors have revised the manuscript carefully and the revised version could be published in the journal.
Decision- Accept
------------------------------------------------------
Congratulations to the authors.
I wish the authors success in their future academic studies.
Kind regards.
Author Response

(The authors gave the same response as above.)
